# Long-Term Visual Outcomes for Small Uveal Melanoma Staged T1 Treated by Proton Beam Radiotherapy

**DOI:** 10.3390/cancers11081047

**Published:** 2019-07-24

**Authors:** Adélaïde Toutée, Martina Angi, Sylvain Dureau, Christine Lévy-Gabriel, Livia Lumbroso-Le Rouic, Rémi Dendale, Laurence Desjardins, Nathalie Cassoux

**Affiliations:** 1Department of Ocular Oncology, Institut Curie, 26, rue Ulm, 75005 Paris, France; 2National Cancer Institute IRCCS Foundation, via Venezian, 1, 20133 Milan, Italy; 3Department of Biostatistics, Institut Curie, 26, rue Ulm, 75005 Paris, France; 4Department of Radiation Oncology, Institut Curie, 26, rue Ulm, 75005 Paris, France; 5School of Medicine Paris V Descartes, PSL University, 12 rue de l'Ecole de Medecine, 75006 Paris, France

**Keywords:** uveal melanoma 1, staged T1 2, proton beam radiotherapy 3

## Abstract

There is increasing evidence of the survival benefit of treating uveal melanoma in an early stage, however it is important to discuss with the patient the associated risk of visual loss. We investigated visual outcomes for uveal melanomas staged T1 (T1UM) treated by proton beam radiotherapy (PBR) as a function of their distance to fovea-optic disc. This retrospective study included a cohort of 424 patients with T1UM treated with PBR between 1991 and 2010 with at least a 5-year follow-up. Visual acuity (VA) was analyzed for patients with posterior edge of tumor located at ≥3 mm (GSup3) or <3 mm (GInf3) from fovea-optic disc. The mean follow-up duration was 122 months, no tumor recurrence was observed. The mean baseline and final VA were 20/25 and 20/32 for GSup3 (*n* = 75), and 20/40 and 20/80 for GInf3 (*n* = 317) respectively. The frequency of a 20/200 or greater visual conservation was 93.2%(CI95%:87.7–99.1) and 60.1%(CI95%:54.9–65.9) for GSup3 and GInf3 respectively. This difference between groups was statistically significant (*p* < 0.001). The risk factors for significant VA loss (less than 20/200) were GInf3 location (*p* < 0.001), tumor touching optic disc (*p* = 0.04), initial VA inferior to 20/40 (*p* < 0.001), documented growth (*p* = 0.002), and age greater than 60 years (*p* < 0.001). In summary, PBR for T1UM yields excellent tumor control and good long-term visual outcomes for tumors located ≥3 mm from fovea-optic disc.

## 1. Introduction

Uveal melanoma (UM) is the most common primary ocular malignancy among adult patients. As successful treatment of medium and large UM does not influence survival, there is increasing interest in early diagnosis and prompt effective treatment of UM at an early stage, with the aim of avoiding liver metastases and a fatal outcome [1]. Ocular radiotherapy, including brachytherapy and teletherapy, has been developed to spare the globe and a potential visual acuity while eradicating the primary tumor. The treatment of choice is selected according to the dimensions and the location of the tumor, the patient’s condition and the technical platform of the oncology center. Conservative treatment of UM with proton beam radiotherapy (PBR) [2,3] or Iodine-125 (I^125^) plaque brachytherapy has been available in our center since 1991.

Visual outcome and prognosis of small choroidal melanoma after treatment with I^125^ plaque radiotherapy were recently described by Shields et al [4]. They reported that small choroidal melanoma can potentially be dangerous, with a 10-year risk for metastasis at 8.8%. Five years after brachytherapy, 38.2% of patients had a vision inferior to 20/200. To our knowledge, no study to date has described visual outcome and prognosis focused on small UM treated by PBR.

Proton beam provides very precise, homogenous, and localized radiation dose distribution to the entire tumor while sparing of normal tissues outside the field. Irradiation of the same volume by I^125^ brachytherapy leads to higher dose at the base and more diffusion of radiation. The calculated irradiation volume with PBR includes the tumor diameter with a safety margin of 2.5 mm around it [3]. Ocular tissue outside of the proton beam is therefore not exposed to radiation.

Visual retention after PBR depends mainly on location and size of the tumor [5,6,7,8]. On one hand, visual outcomes depend on tumor location, as tumors located close to the macula or optic disc develop radiation maculopathy and neuropathy thus causing severe visual loss. On the other hand, visual outcomes depend on the irradiated volume. For large irradiated tumors, the toxic tumor syndrome can be responsible for exudation and inflammation leading to visual loss [9]. Previous reports, even large series, have included tumors of different sizes as well as different locations [10].

In our study, we focused only on small posterior UM staged T1 (T1UM) to evaluate the visual results after PBR. We compared 2 groups of patients affected by a T1UM. We defined these two groups according to the position of the posterior edge of the tumor from the fovea and optic disc with a threshold of 3 mm.

The aim of this study was to present the long-term visual outcomes as a function of tumor location for T1UM treated by PBR, in order to provide information that will allow patient counselling when discussing risks and benefits of early treatment of small UM.

## 2. Materials and Methods

Institutional Review Board of the Institut Curie approval was obtained for this study. This study followed the tenets of 8th Declaration of Helsinki.

Since 1983, all the patients that have been treated at the Institut Curie for UM are registered prospectively in database MACRO version 3.0.87 Inframed limited 2013. For this retrospective study, we analyzed the data from all consecutive patients treated by PBR in our center from November 1st, 1991 to December 31, 2010. Each case was classified according to the guidelines of the 8th edition of the American Joint Committee on Cancer Staging system. T1UM are defined as choroidal melanoma or ciliary body melanoma with tumor base ≤ 9 mm and thickness ≤ 6 mm or with tumor base ≤ 12 mm and thickness ≤ 3 mm, with no extraocular extension of more than 5 mm [11]. All T1UM patients with at least 5 years of follow-up were included in this study. Exclusion criteria were iris melanoma, infiltrating melanoma, and ring ciliary body melanoma.

Full ophthalmologic examination was performed at initial examination with visual acuity (VA) using Snellen chart, ocular pressure, slit lamp biomicroscopy, and undirected ophthalmoscopy. Patients underwent fundus photography, B-scan ultrasonography, and optical coherence tomography (OCT). Diagnosis was based on clinical features and ultrasonographic appearance, no biopsy was performed. Clinical features of the choroidal tumor were tumor dimensions (expressed in millimeters), orange pigment, subretinal fluid, and association with a nevus of Ota or not. Documented growth, extrascleral spread, initial retinal detachment, initial macular edema, and intravitreal hemorrhage were registered. They underwent orbital magnetic resonance imaging for juxta papillary choroidal lesion. Tumor thickness was measured by B-scan ultrasonography. Basal diameter was measured on fundus photographs and by transillumination and the largest measurement was selected.

The treatment planning PBR protocol for UM has been described previously by Dendale and Desjardins et al [2,3], using the treatment planning system Eyeplan. A dose of 60 Gray RBE (relative biological effectiveness) in 4 fractions was delivered after surgical clip positioning. No patient in this cohort received pharmacologic therapy for radiation vasculopathy during follow-up.

Patients had been followed longitudinally for ocular outcomes, with an ophthalmologic examination at the Curie Institute one month after treatment, then every six months for 3 years, and at 4 years, 5 years, and 10 years after treatment.

Visual outcomes were analyzed according to the distance between the posterior edge of the tumor and the fovea-optic disc, by comparing two groups of patients: one with a distance superior or equal to 3 mm (GSup3) and the other with a distance inferior to 3 mm (GInf3).

Quantitative data were described as means and range, respectively. Qualitative data were presented as numbers and proportions in statistical analysis. The overall survival was defined as the time from diagnosis to the date of last follow-up. Survival and median follow-up were estimated by Kaplan Meier method. Curves of VA conservation after PBR were also computed using Kaplan–Meier method. The delay to VA loss is defined as the time from PBR to the time when the VA crossed the thresholds—20/200 or 20/40 Snellen. Patients already under the threshold of VA at the initial time of PBR were not taken into account in the curves of VA conservation. A Log-Rank test was used to compare in univariate analysis groups according to distance to the fovea-optic disc. A multivariate analysis using a Cox regression model was done to determine factors influencing the loss of VA. All factors available were included in the model. Univariate and multivariate analyses used the threshold of 20/200 as event in order to keep in in the analyses the most patients. Statistical analysis was performed using R software version 3.2.2.

Comparison between groups was done in the univariate analysis using the Log-Rank test and multivariate regression used the VA threshold of 20/200, which takes into consideration most patients at the initial time of PBR. Statistical analysis was performed using R software version 3.2.2.

## 3. Results

Among the 8399 UM patients recorded in our database, 594 were T1UM with at least 5 years of follow-up and none of the exclusion criteria. Of these, 170 were treated by brachytherapy, hence 424 T1UM patients fulfilled the inclusion criteria and were included in this retrospective study.

The median follow-up was 10.2 years (range, 5.1–23 years). Overall, the mean age at initial diagnosis was 56.2 years (range, 14–88 years). There were 220 women (51.9%) and 204 men (48.1%). The median largest tumor diameter was 8.8 mm (range, 3.0–12.0mm) and the median largest tumor height was 2.8 mm (range, 0.7-5.8mm). Only one patient had a tumor staged T1d with an extraocular extension, which was less than 5 mm in diameter. Documented growth of tumor was observed in 141 patients (33.3%) and was the reason for treatment of smaller lesions. Only forty-five tumors came into contact to the optic disc. Patient demographic data and initial tumor characteristics are described in Table 1.

The overall survival rate was 91.7% (95%CI, 88.6–94.9%) at 10 years. The patient status lifespan is detailed in Table 2. No ocular recurrence was observed during the follow-up period. A total of 39 patients died within 5 years of follow-up. The cause of death was UM metastatic disease in 9 patients (23.1%), another cancer in 8 patients (20.5%), an intercurrent disease in 3 patients (7.7%), and unknown causes in 19 patients (48.7%).

### 3.1. Visual Acuity Outcomes

Visual outcomes after PBR were analyzed for the cohort and for the two groups (GSup3, GInf3) as a function of the distance between the posterior edge of the tumor and the optic disc or the fovea.

#### 3.1.1. Visual Outcomes in the Cohort

At the time of PBR, 117 patients had a best corrected visual acuity (BCVA) lower than 20/40, and 16 patients had a VA lower than 20/200. The mean initial BCVA was 20/40. Visual conservation was illustrated by the Kaplan–Meier curves estimating the time from the PBR and the moment of reaching the thresholds of 20/40 (Figure 1) or 20/200 (Figure 2).

The probability of visual conservation of 20/40 or better since the PBR was 79.5% (CI95%: 75.1–84.1) at 2 years, 48.2% (CI95%: 42.9–54.1) at 5 years and 35.5% (CI95%: 30.2–41.8) at 10 years. The probability of patients retaining vision of 20/200 or better was 87.0% (CI95%: 83.8–90.3) at 2 years, 65.9% (CI95%: 61.5–70.7) at 5 years, and 52.8% (CI95%: 47.7–58.4) at 10 years.

#### 3.1.2. Visual Outcomes in Gsup3 and in GInf3

Among the 75 patients of GSup3, the mean BCVA was 20/25 at baseline and 20/32 five years after treatment. 

GInf3 included 317 patients. The mean baseline and final BCVA were 20/40 and 20/80 respectively.

When the conservation of a visual acuity of 20/200 or better was assessed according to the distance to fovea-optic disc (Figure 3), a significant difference between the two groups was observed. The frequency of a maintaining a 20/200 or greater visual acuity since the PBR in GSup3 was 97.3% (CI95%: 93.7–100) at 2 years, 93.2% (CI95%: 87.7–99.1) at 5 years, and 82.7% (CI95%: 72.3–94.6) at 10 years. The frequency of a 20/200 or greater visual conservation in GInf3 was 85.3% (CI95%: 81.4–89.4) at 2 years, 60.1% (CI95%: 54.9–65.9) at 5 years, and 47.0% (CI95%: 41.3–53.5) at 10 years. The cumulative rates of visual acuity conservation (20/200 or better) were statistically better for Gsup3 compared to Ginf3 during the entire follow-up (p < 0.001), as illustrated in Figure 3.

### 3.2. Risk Factors Associated with Vision Loss after Irradiation

To assess the factors associated with the loss of VA under the threshold of 20/200, a multivariate analysis was performed using a cox regression model. Variables which integrate the model were available at the time of PBR, as shown in Table 3. Risk factors of vision loss inferior to 20/200 were the distance to fovea-optic disc inferior to 3 mm (GInf3) (*p* < 0.001), tumor touching optic disc (*p* = 0.04), initial VA inferior to 20/40 (*p* < 0.001), documented growth (*p* = 0.002), and age older than 60 years (*p* < 0.001). Primary retinal detachment was not a risk factor of vision loss (*p* = 0.55). Therefore, the localization of the tumor influences the risk of vision loss.

## 4. Discussion

Our study presents the long-term visual outcomes for a large longitudinal cohort of small T1UM treated by proton beam radiotherapy in a tertiary oncology care center. The localization of the tumor in relation to the distance from the fovea-optic disc influences the risk of vision loss. 

The risk of vision loss related to side effects of radiotherapy is a source of anxiety and a recurrent question from patients at the first consultation. PBR has proved efficiency to achieve local control tumor in T1UM, to spare the globe and maintain a useful vision [2,3,12,13,14,15,16,17].

Our retrospective analysis of 424 small T1UM treated with PBR and a long-term follow-up demonstrated favorable visual outcomes. Indeed, patients with a baseline visual acuity of 20/40 or more retained the same vision after a minimum of 5 years after irradiation in 48.2% (CI95%:42.9–54.1) of patients. Similarly, 65.9% (CI95%: 61.5–70.7) of patients with a baseline VA superior to 20/200 retained this vision at 5 years. 

Many studies describe visual outcomes after PBR of UM, according to tumoral volume [10] or according to involvement of the fovea and/or optic disc [5,6,18,19]. Visual acuity is generally preserved in patients with smaller UM located further from the posterior pole [6,18,19,20,21]. However, our study is the first which focuses specifically on visual outcomes of T1UM treated by PBR according to the proximity of the tumor to visually important structures. The threshold of 3 mm was defined to free itself from the irradiated surface with the 2.5 mm security margin used for the delivery of the proton beam.

Factors of poor vision outcome of 20/200 or worse for T1UM treated by PBR were: tumor located at less than 3 mm from fovea or optic disc (*p* < 0.001), touching the optic disc (*p* = 0.04), with a documented growth (*p* = 0.002) or patients with an initial VA inferior to 20/40 (*p* < 0.001) and older than 60 years (*p* < 0.001). 

A previous study indicated that factors related to a visual outcome of 20/200 or worse after proton beam or plaque radiotherapy were greater tumor thickness, submacular fluid, closer proximity to optic disc and foveola, poor pretreatment vision, and increasing radiation dose to optic disc, fovea, and lens [22]. One other study reported visual outcome according to a defined distance of 5 mm between UM and the posterior pole, but after treatment by plaque radiotherapy. Visual outcomes were worst for UM located at less than 5 mm to fovea or optic disc, with a visual acuity inferior to 20/200 in 35% at 5 years versus 25% for UM located at more than 5 mm [23]. 

Papakostas et al [10] described long-term outcomes for only large choroidal melanomas after PBR. The study also showed that vision loss was dependent on the proximity of the tumor to optic disc and fovea. 

In our analysis, the localization of the tumor also influences the risk of vision loss with a statistically lower visual conservation for small tumors near the fovea-optic disc (*p* < 0.001). Patients of GSup3, with a tumor located far away from the main visual structures, retained an VA superior to 20/200 in 93.2% (CI95%:87.7–99.1) at 5 years. Patients of Ginf3, with a tumor near fovea-optic disc, had an initial and a final BCVA lower than GSup3 and retained a VA superior to 20/200 in 60.1% (CI95%:54.9–65.9) at 5 years (*p* < 0.001). Tumor touching the optic disc is also a risk factor of severe vision loss probably due to the optic neuropathy induced by the radiotherapy [7,8,9]. 

For tumors close to the fovea, a direct toxicity induced by the radiation dose to the vascular endothelium leads to macular ischemia and is a risk factor of vision loss [7,18,24,25]. The risk of radiation maculopathy is related to the radiation dose and to the volume of macula irradiated. Nevertheless, patients with a tumor near the posterior pole will not have an inevitable severe vision loss [5,18]. Indeed, over half (60.1%) of GInf3 retained a VA superior to 20/200 five years after PBR in our study. Patel et al [18] reported that many patients with choroidal melanoma having foveal involvement, without tumor size distinction, maintained a useful vision despite receiving a full dose of proton radiation to the fovea. They reported that more than 35.5% patients retained 20/200 or better 5 years after PBR among 351 patients. For those patients with a baseline BCVA of 20/40 or better, 16.2% of patients retained this level of vision 5 years after PBR. They found two factors associated with a better visual outcome: tumor height less than 5 mm and baseline visual acuity of 20/40 or better (*p* < 0.001).

Visual outcomes also depend on the irradiated volume. A smaller irradiation volume should reduce the radiation toxicity. Recently, Shields et al [4] described visual outcome for 1780 small choroidal melanoma treated by plaque radiotherapy. With this different conservative radiotherapy, the rates of visual acuity inferior to 20/200 were 38.2% (CI95%: 35.5–41.1) at 5 years [4], which is relatively similar to our results.

Preserving visual acuity is an important aspect in the management of small choroidal melanoma in the posterior fundus. Intravitreal injections were not usually performed ten years ago in prevention or treatment of radiation maculopathy. Nowadays, the hope is that long-term visual outcomes will be better with new complementary therapy like intravitreal injection of anti-vascular endothelium growth factor (anti-VEGF) or dexamethasone. Thanks to the arrival of OCT-angiography, an earlier radiation maculopathy detection is possible [25] and maculopathy treatment could be given before clinical maculopathy [26]. Further prospective studies are needed to evaluate visual outcomes with earlier complementary treatment given before clinical radiation complication.

Regarding vital prognosis, the overall survival rate was relatively good with 91.7% (95%CI, 88.6–94.9%) at 10 years but T1UM can be a potentially life-threatening disease with 2.8% of metastases at 5 years. Shields et al [4] described prognosis for small choroidal melanoma treated by plaque radiotherapy. They had a risk of metastases of 4.5% (95%CI: 3.4–5.9%) at 5 years, similar to ours. Small choroidal melanoma required an early treatment in order to prevent increase in melanoma thickness. The documented growth of the tumor in thickness or diameter is a predictive factor of metastasis of UM, each millimeter increase in thickness showed a 1.06 hazard ratio of melanoma metastasis [27].

All tumor met international criteria of uveal melanoma (AJCC 8th) staged T1 [11]. Uveal melanoma is one of the few tumors for which the treatment is initiated without cytopathological diagnosis. In our study, tumor diagnosis was clinical, and no patient had a fine-needle aspiration biopsy (FNAB) in diagnostical process and prognosis in our cohort. The practice of FNAB was not a current practice ten year ago.

The difference in the number between both groups was expected for UM staged T1 [28]. GInf3 has a higher population than GSup3, 317 and 75 patients respectively, due to a probably easier diagnosis when the tumor is located near posterior pole. Tumors near posterior pole cause earlier symptoms like visual loss (due to serous retinal detachment or due to a direct injury of fovea or optic disc) and were visible during the non-mydriatic fundus. Small tumors of GSup3 are more difficult to diagnose and would be found with a systematic dilatation fundus examination for symptoms like photopsia or during incidental findings.

Our study has demonstrated that T1UM located away from the posterior pole can be treated by proton beam with a limited risk of long-term visual loss. 

Traditionally, ophthalmologists tended to observe suspicious melanocytic choroidal lesion near the posterior pole because of the fear of ultimate visual loss from radiation and accepted the risk of documented growth of malignant lesion. More recently, the goal is shifting towards identifying melanocytic lesions that are likely to metastasize and ablate them before they do so [29]. In selected cases, PBR treatment could be used for very suspicious melanocytic choroidal lesions away from the posterior pole with at least 3 risk factors of tumor growth [27,28,30,31,32,33], especially if the patient demands treatment and is well informed of the benefits and risks. These choroidal lesions, if located more than 3 mm from fovea and optic disc, can be treated with PBR with no risk of visual loss.

## 5. Conclusions

Our study confirms that patients with T1UM can retain good vision after conservative treatment by PBR, enhancing our ability to counsel patients on the risk and benefit of early treatment of small melanomas. The location of the tumor influences the risk of vision loss. Long-term visual outcomes are excellent for tumors located the furthest away from posterior pole. Visual results are acceptable if posterior tumor edge is less than 3 mm from fovea and optic disc. The risk of poor vision outcome of 20/200 or worse is highest for tumors located at less than 3 mm from fovea or optic disc, touching optic disc, that have a documented growth or for patients with an initial VA inferior to 20/40 and for patients older than 60 years of age.

## Figures and Tables

**Figure 1 cancers-11-01047-f001:**
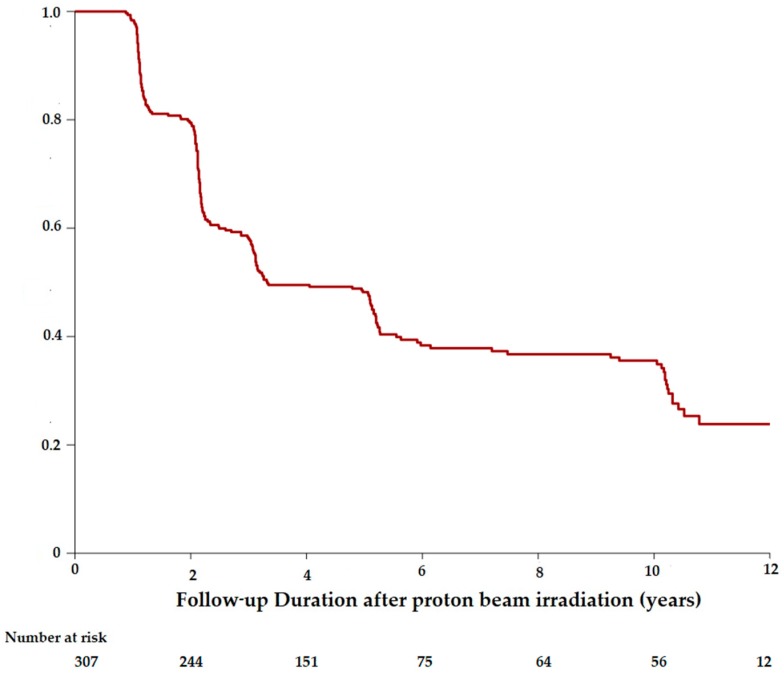
Kaplan–Meier estimates showing the proportion of patients maintaining a visual acuity of 20/40 or better over time, among the patients with a uveal melanoma staged T1 treated by proton beam radiotherapy.

**Figure 2 cancers-11-01047-f002:**
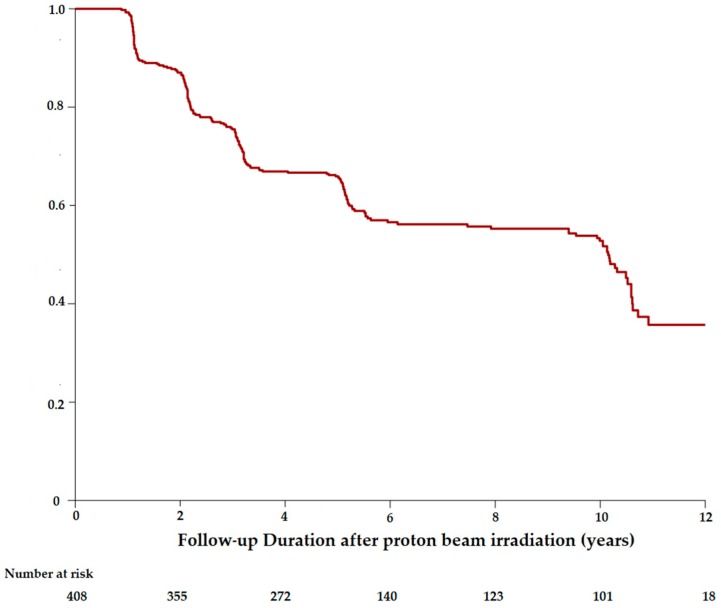
Kaplan–Meier estimates showing the proportion of patients maintaining a visual acuity of 20/200 or better over time, among the patients with a uveal melanoma staged T1 treated by proton beam radiotherapy.

**Figure 3 cancers-11-01047-f003:**
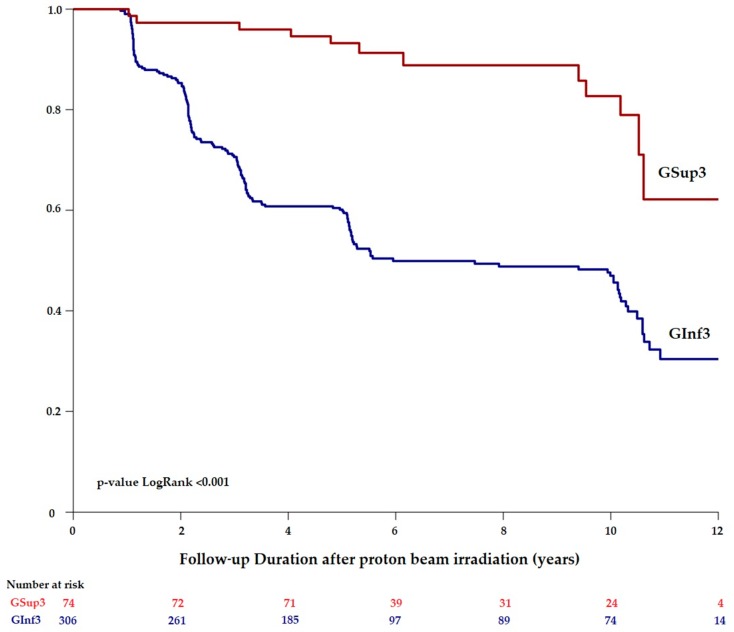
Conservation of a visual acuity of 20/200 or better over time for patients with a uveal melanoma staged T1 as a function to their distance to the fovea-optic disc superior or equal to 3 mm (GSup3) or inferior to 3 mm (GInf3). Comparison using Log-Rank test (*p*-value < 0.001).

**Table 1 cancers-11-01047-t001:** Uveal melanoma staged T1 treated by proton beam radiotherapy with 5-year follow-up in 424 patients: Patient demographic characteristics and baseline tumor characteristics.

Characteristics	Value	Range or Percentage
Sex, no., %		
	Female	220
	Male	204
Eye, no., %		
	Right	226
	Left	198
Tumor dimension, median (range), mm		
	Basal tumor diameter	8.8
	Tumor thickness	2.8
Distance, mean (range), mm		
	To optic disc	3.3
	To fovea	2.6
Ocular location of the tumor, no., %		
	Tumor invaded ciliary body	6
	Tumor touching optic disc	45
	Non applicable	373
Ocular location of the tumor, no., %		
	Ante equatorial location	11
	Equatorial location	30
	Post equatorial Location	383
Documented growth, no., %	141	33.3%
Initial Findings, no., %		
	Initial Extra scleral spread	1
	Initial retinal detachment	54
	Initial macular edema	15
	Intravitreal hemorrhage	2
	Amelanotic tumor (partially or totally unpigmented)	73

**Table 2 cancers-11-01047-t002:** Uveal melanoma (UM) staged T1 treated by proton beam radiotherapy (PBR) with 5-year follow-up in 424 patients: Patient status, lifespan, and evolution.

Characteristics	Patients	Percentage
Patient status, no., %		
	Alive	385
	Dead from all-cause mortality	39
T1UM with metastatic disease, no., %		
	Initial UM metastasis	0
	Patients with UM metastases during 5-year follow-up	12
	Patients without metastases	412
Metastases location, no., %		
	Liver metastases	11
	Multisystemic metastases (liver, lungs, skin)	1

**Table 3 cancers-11-01047-t003:** Risk Factors associated with vision loss inferior to 20/200 after irradiation by proton beam radiotherapy (*n* = 424).

	Patients	Hazard Ratio	95% Confidence Interval (95%CI)
**Age (years)**			
≤60	243	1	
>60	165	1.75	(1.28; 2.38)
**Documented growth**			
Yes	133	1	
No	275	0.6	(0.43; 0.83)
**Distance to fovea-optic disc**			
<3 mm (GInf3)	306	1	
≥3 mm (GSup3)	74	0.29	(0.16; 0.54)
**Initial visual acuity (Snellen scale)**			
<20/40	155	1	
≥20/40	250	0.45	(0.33; 0.62)
**Tumor touching optic disc**			
No	359	1	
Yes	42	1.62	(1.06; 2.49)
**Initial retinal detachment**			
No	355	1	
Yes	52	0.87	(0.55; 1.38)

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
