# Peer review of "Long-Term Visual Outcomes for Small Uveal Melanoma Staged T1 Treated by Proton Beam Radiotherapy"

_cancers, 2019, doi:10.3390/cancers11081047_

Round 1

Reviewer 1 Report

The review  by A. Toutée and collaborators  deals with benefits, outcomes and survival of patients suffering frorm Uveal Melanoma. The text is well written ans does not deserve major edition ; the data are clearly presented. I have no major concerns  with this large range anlaysis  and do not request further changes before consideration of publication. 

Author Response

Thank you very much for the positive feedback

Reviewer 2 Report

The study design with selection of a subpopulationof 424 patients with a T1UM tumour within the larger cohort with UM (8399 patients) should be better motivated and described. 

Comparison not only between the two groups of patients with different positions of T1 tumour but also as a general reference to the whole material may be presented.

The figure 1 is missing from the generated pdf and cannot be judged. Must be included.

Text revision of sentence on line 160-161 is needed.

Typos should be checked ie line 215 - closed should read close.

Author Response

Thank you very much for your comments and suggestions. 

- Point 1: The study design with selection of a subpopulationof 424 patients with a T1UM tumour within the larger cohort with UM (8399 patients) should be better motivated and described. 

Response 1: The selection of the subpopulation was done in accordance with the inclusion and exclusion criteria detailed in the material and methods. We have clarified this in the manuscript:

"Among the 8399 UM patients recorded in our database, 594 were T1UM with at least 5 years of follow-up and none of the exclusion criteria. Of these, 170 were treated by brachytherapy, hence 424 T1UM patients fulfilled the inclusion criteria and were included in this retrospective study. "

- Point 2: Comparison not only between the two groups of patients with different positions of T1 tumour but also as a general reference to the whole material may be presented.

Response 2:  Visual acuity was analysed and illustrated in Kaplan-Meier analysis both for the whole cohort of T1UM (Figure 1 and 2) and for the two groups according to the distance from fovea and optic disc (Figure 3). We preferred to perform separate Kaplan-Meier analyses and we didn't overlay Figure 2 and 3 to simplify comprehension.

-Point 3: The figure 1 is missing from the generated pdf and cannot be judged. Must be included.

Response 3: We apologise for the accidental omission of Figure 1: it has now been embedded in the manuscript.

-Point 4: Text revision of sentence on line 160-161 is needed.

Response 4: Thank you for the kind advice. The text has been revised as follow: "The cumulative rates of visual acuity conservation (20/200 or better) were statistically better for Gsup3 compared to Ginf3 during the entire follow-up (p<0.001), as illustrated in Figure 3."

-Point 5: Typos should be checked ie line 215 - closed should read close

Response 5: Thank you for your correction. English language of the entire manuscript has been edited 

Reviewer 3 Report

In this retrospective work, the authors aimed at assessing the visual outcomes in uveal melanoma after proton beam therapy, as a function of their distance to fovea/optic disc. The main results demonstrated that the location of tumor mass influences the risk of visual loss. In addition, proton beam therapy is an effective conservative treatment in controlling tumor mass, and singifinatly improves long-term visual outcomes. When evaluated as a whole, this work is adequately performed and quite well written. 

However, i have serious concerns about the novelty of this study. Despite the authors focused this retrospective study on small uveal melanoma (stage 1), there are a great number of reports already published in literature demonstrating that proton beam therapy is beneficial in terms of tumor mass control and visual outcomes.

If the authors believe their work is particularly innovative for some specific reasons, they should better highlight these aspects, and clearly indicate the need of another retrospective study like this.

Author Response

Response: Thank you for the positive feedback on the overall quality of our work. As noted, there are a number of previous publications on proton beam, however none is focused on only small tumours. In an era when more and more attention is being drawn to the importance of early diagnosis and treatment of small UM in the hope to positively influence survival, preservation of visual acuity remains the main concern. Our large retrospective study offers solid data on long-term visual results, offering a 3-mm threshold that can be easily be applied when counselling the patient. As suggested by the Reviewer, we have modified both the introduction and the discussion to better highlight these aspects.

Round 2

Reviewer 3 Report

The authors carefully revised the manuscript according to the reviewers' suggestions, and provided exhaustive explanations.